# ESCAPE: Equivariant Shape Completion via Anchor Point Encoding

## Abstract

Shape completion, a crucial task in 3D computer vision, involves predicting and filling the missing regions of scanned or partially observed objects. Current methods often suffer from orientation-dependent inconsistencies, particularly under varying rotations, limiting their real-world applicability. We introduce **ESCAPE** (Equivariant Shape Completion via Anchor Point Encoding), a novel framework designed to achieve rotation-equivariant shape completion. Our approach employs a distinctive encoding strategy, representing objects by selecting anchor points and utilizing them in a distance-based encoder akin to the D2 shape distribution. This enables the model to capture a consistent, rotation-equivariant understanding of the object's geometry. ESCAPE leverages a transformer architecture to encode and decode the distance transformations, ensuring that generated shape completions remain accurate and equivariant under rotational transformations. Additionally, we perform optimization to refine the predicted shapes from anchor point positions and predicted encodings, Experimental evaluations demonstrate that ESCAPE achieves robust, high-quality reconstructions across arbitrary rotations and translations, showcasing its effectiveness in real-world applications.

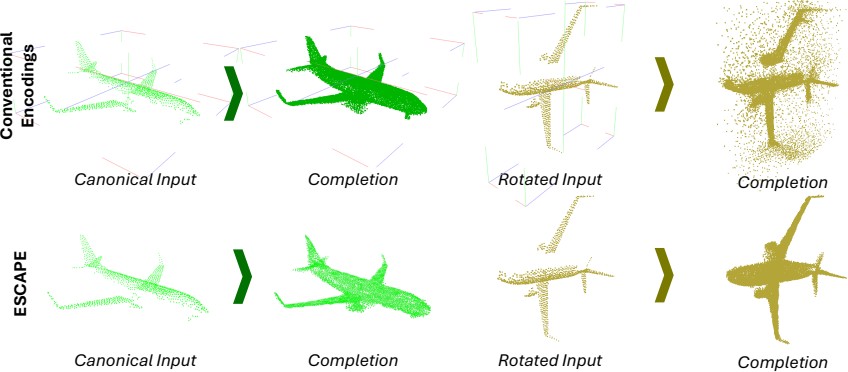

Figure 1: Existing shape completion methods (Top) use conventional canonical coordinates and perform poorly under rotation changes and unknown canonical reference. Using our anchor point encoding (Below), we manage to consistently complete the shape with arbitrary rotation in non-canonical coordinates.

## 1 Introduction

3D perception has been dependent on canonical orientations, causing significant challenges in dynamic environments, such as robotic manipulation or real-time object recognition, where objects interact from varying viewpoints. Traditional methods, like voxel-based approaches and early point cloud networks such as PointNet(21) and PointNet++(22), have laid a strong foundation but cannot inherently handle rotational variance. Recent methods have introduced improvements by leveraging attention mechanisms and hierarchical processing capabilities but still rely on orientation

normalization or data augmentation techniques to manage rotations, which do not fully resolve the inherent challenges of varying viewpoints.

Despite progress in the field, existing shape completion techniques struggle to maintain robustness and accuracy under arbitrary rotations. This notably limits their practicality in real-world applications where objects can appear in any orientation. Recent methods, including transformer-based models like SnowflakeNet(35) and SeedFormer(46), have introduced improvements by leveraging the attention mechanisms, progressively growing points, and hierarchical processing capabilities of transformers. However, these models expect partial shapes in aligned orientation across a category and only learn completion in canonical coordinates, which do not guarantee performance consistency in varying viewpoints or unknown local coordinates.

In addition, progress has been made in developing rotation-invariant descriptors, particularly in shape registration tasks, which require detecting points of interest in objects regardless of orientation. However, these advancements have yet to be fully extended to shape completion. These descriptors are vital for applications such as 3D model retrieval and pose estimation but do not directly facilitate the reconstruction of entire object geometry in a rotation-equivariant manner.

To bridge these gaps, we introduce ESCAPE, a novel approach to ensure rotation-equivariant shape completion. ESCAPE employs a unique encoding strategy by selecting high-curvature anchor points around the input shape and transforming point coordinates into a set of distances from these anchor points. This method constructs a robust representation of objects equivariant to their orientation. The representation is processed through a transformer architecture, which enables the model to reconstruct the distance to the same anchor points on a complete object point set under varying conditions. Subsequently, an optimization is utilized to find point coordinates of the completed geometry from the predicted distances.

We propose a complete pipeline to generate, encode, decode, and interpret rotation equivariant features for the first non-canonical 3D shape completion.

In summary, our contributions are threefold:

- We introduce ESCAPE, a novel rotation-equivariant 3D encoding strategy using high-curvature anchor points, enabling robust shape description and reconstruction.
- We develop a transformer-based architecture that leverages our equivariant encoding to generate the completed point cloud, maintaining consistency across varying object orientations and partial inputs.
- We present the first end-to-end rotation-equivariant shape completion method, demonstrating robust performance under arbitrary rotations and in the absence of canonical object coordinates.
- We establish and rigorously evaluate a challenging real-world shape completion benchmark using the OmniObject dataset(34), featuring partial point clouds with diverse geometries and arbitrary poses.

## 2 RELATED WORKS

### 2.1 POINT CLOUD PROCESSING

Point clouds, representing unordered and sparse 3D data, challenge conventional CNNs. Voxel-based methods like VoxNet (17) and VoxelNet (47) convert point clouds into regular grids for 3D CNN processing but are computationally intensive. PointNet (21) and PointNet++ (22) address unordered sets with MLPs and hierarchical grouping. PointNet creates a permutation-invariant structure with pointwise MLPs, while PointNet++ adds hierarchical layers and furthest point sampling, increasing computational complexity. These methods are often combined with voxel-based approaches in large-scale applications (28; 47).

Graph neural networks (GNNs) like PointCNN (16) and EdgeConv (33) capture local geometric features through spatial relationships and dynamic graph construction. Recent advancements integrate edge-based attention for improved efficiency (31; 30). Transformers and attention mechanisms have also been applied to point clouds, with Graph Attention Networks (GATs) (31) using self-attentional

layers and the Transformer architecture (30) introducing multi-head attention for robust feature aggregation. Set Transformer (15) and Perceiver (13) adapt attention mechanisms for 3D data. Point cloud-specific transformers, such as PCT (10) and Point Transformer (45), apply self-attention to neighborhood points, though they can be computationally expensive.

## 2.2 ROTATION-INVARIANT ENCODINGS

Handcrafted rotation-invariant descriptors have been widely explored in 3D by researchers before the popularity of deep neural networks. To guarantee invariance under rotations, many handcrafted local descriptors (25; 24; 29; 11) rely on an estimated local reference frame (LRF), which is typically based on the covariance analysis of the local surface, to transform local patches to a canonical representation. The major drawback of LRF is its non-uniqueness, making the constructed rotational invariance fragile and sensitive to noise. Consequently, attention has shifted to LRF-free approaches (5). These methods focus on mining the rotation-invariant components of local surfaces to represent the local geometry. For instance, PPF (5), PFH (25), and FPFH (24) encode the geometry of the local surface using histograms of pairwise geometrical properties. Despite being rotation-invariant by design, these handcrafted descriptors are often inadequate for complex geometry and noisy data.

Recently, many deep learning-based methods have aimed to learn rotation-invariant descriptors. PPF-FoldNet (4) encodes PPF patches into embeddings, using a FoldingNet (37) decoder to reconstruct the input, enabling correspondences from the rotation-invariant embeddings. SpinNet (1) and Graphite (26; 27) align local patches to defined axes before learning descriptors. However, these methods are limited by their locality, as descriptors are learned only from the local region, making them less distinctive.

YOHO (36) introduces a rotation-equivariant approach by leveraging an icosahedral group to learn a group of rotation-equivariant descriptors for each point. Rotational invariance is achieved by max-pooling over the group, but this method struggles with efficiency and complete rotational coverage. Object-centric registration methods (38; 6) strengthen rotational invariance by combining rotation-invariant descriptors with rotation-variant inputs, though performance drops under large rotations.

In point cloud classification, methods (44; 3) describe whole shapes as rotation-invariant descriptors but lack global awareness in node/point descriptors. Methods such as Transformer-based approaches (30) aim to incorporate global context, but robustness to rotational changes is often achieved through data augmentation, which is not optimal. Recent works such as Rotation-Invariant Transformer (40) and Riga (39) have proposed more robust rotation-invariant and globally-aware descriptors. Despite these advancements, these approaches have not yet been thoroughly explored in shape completion tasks, highlighting a significant area for future research.

## 2.3 SHAPE COMPLETION METHODS

Pre-deep learning point cloud completion methods rely on object symmetry (18), or a database of complete shapes (20) to achieve effective results. However, these methods are constrained by the need for specific preconditions to be met by the input data, limiting their applicability compared to deep learning-based approaches.

Learning-based methods can be classified into two primary categories based on the type of representation used for 3D data: methods utilizing point clouds and methods employing alternative representations such as voxel grids, implicit functions, and others. Methods using alternative representations often need higher memory consumption, making them less scalable for high-resolution inputs. Although researchers have proposed more efficient representations such as Octrees (32), sparse lattice networks (23), and sparse convolution operations (9), none have proven to be as efficient and effective as directly processing 3D point coordinates.

PointNet (21) and its variant PointNet++ (22) introduced specialized operations that enable learning directly from point cloud coordinates, revolutionizing point cloud processing tasks. PCN (43) was built on top of PointNet and became the first deep learning-based method for point cloud completion, utilizing an encoder-decoder architecture with a folding operation to complete given partial input point clouds. Following PCN, many other methods with similar model architectures have been developed (43). In addition to these methods, some researchers formulate point cloud completion

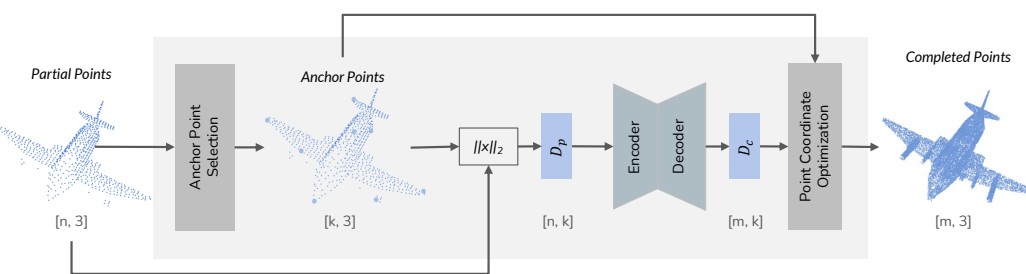

Figure 2: The overall pipeline for ESCAPE model. Initially, it extracts $a$ anchor points to construct rotation-invariant features as input to a transformer-based encoder-decoder architecture. The transformer is specially modified to predict the distance between points in the complete geometry and the extracted anchor points. It simultaneously constructs complete geometry with $m$ points and predicts the distance to anchor points. Finally, an optimization procedure has been utilized to find the coordinates of the complete shape.

as a probabilistic problem, where a given input cloud can be mapped to multiple complete point clouds. They address point cloud completion by introducing probabilistic methods such as Variational Autoencoders (14) and GANs (8).

Another milestone for point cloud completion is the application of transformer-based methods. Due to their efficiency with unordered data, transformers are well-suited for point cloud processing tasks. PoinTr (41) formulates the completion task as set-to-set translation, utilizing transformers for the first time in this context. SnowflakeNet (35) employs an encoder-decoder architecture based on transformers to complete point clouds in a coarse-to-fine fashion. Each level of the decoder architecture inputs a prediction of a subset of the complete point cloud generated by the previous level and splits the input into child points to generate a finer prediction. Similarly, Seedformer (46) introduces a new representation called patch seeds, consisting of points and their features, to be used in the decoding step. Unlike SnowflakeNet, Seedformer's decoder levels utilize point features while generating a new set of complete point clouds. AdaPointTr (42) builds on top of PoinTr by incorporating a denoising task and an adaptive query generation mechanism. Finally, our method show some similarities to a recent method named: AnchorFormer (2) which also utilizes points to aid shape completion. It employs points(anchors) around the shape to capture regional information of objects. These points are then used to reconstruct fine-grained object using a modulation scheme.

Although these models predict completion well, they rely on aligning objects into canonical coordinates and fail when local coordinates are unknown, limiting their effectiveness in more generalized and unstructured scenarios. This reliance on known local coordinates and failure to predict under arbitrary rotations highlights a significant challenge that our work aims to address.

## 3 METHODOLOGY

Our method addresses point cloud completion by leveraging a combination of rotation-invariant features and a specially designed transformer model. Initially, anchor points and rotation-invariant features are extracted from the partial input point cloud. These features are then passed into the transformer to predict the complete geometry of the object relative to the anchor points.

It is important to note that the final point coordinate optimization is influenced by the input anchor points, which are affected by the rotation of the input. As a result, the final predictions are also rotated, ensuring that the entire pipeline remains rotation-equivariant. This means that the output is consistently aligned with the input rotation.

The overall architecture is shown in Figure 2. Below, we detail the different components of our ESCAPE model.

## 3.1 ANCHOR POINTS ENCODING

Given a partial point cloud $P = \{p_1, p_2, \ldots, p_n\}, p_i \in \mathbb{R}^3$, we select a set of anchor points $A = \{a_1, a_2, \ldots, a_k\}, a_j \in \mathbb{R}^3$. The distances between points in $P$ and anchor points in $A$ are computed and stored in a distance matrix $D_p \in \mathbb{R}^{n \times k}$, where each element $d_{ij}$ is:

$$d_{ij} = \|p_i - a_j\|_2, \quad \forall i \in \{1, \ldots, n\}, \forall j \in \{1, \ldots, k\}.$$

This distance matrix $D_p$ serves as the input feature set for our transformer model.

The selection of anchor points is crucial. They should be well-distributed and consistent across samples within the same object category. We use furthest point sampling (FPS) to form $k$ clusters to achieve this. Within each cluster, we compute the Laplacian $\Delta$ of the normal vectors $N = \{n_1, n_2, \ldots, n_m\}$ for each point $p_i$ in the cluster:

$$\Delta n_i = n_i - \frac{1}{|\mathcal{N}(i)|} \sum_{j \in \mathcal{N}(i)} n_j,$$

where $\mathcal{N}(i)$ denotes the neighbors of $p_i$.

We estimate PCA-based curvature $\kappa_i$ at $p_i$, defined as the smallest eigenvalue of the covariance matrix $C_i$ of the neighboring normals (12):

$$C_i = \frac{1}{|\mathcal{N}(i)|} \sum_{j \in \mathcal{N}(i)} (n_j - \bar{n})(n_j - \bar{n})^T, \quad \kappa_i = \min(\text{eig}(C_i)),$$

where $\bar{n}$ is the mean normal vector. The highest curvature points within each cluster are selected as anchor points, representing salient landmarks consistent across geometrically similar samples.

## 3.2 TRANSFORMER ARCHITECTURE

Our point cloud transformer model is inspired by AdaPoinTr (42), but we revised it to meet the specific requirements of rotation invariance.

First, we modify the feature extraction process used to generate point proxies. In the original architecture, a DGCNN (33) model is employed to extract local neighbor features through hierarchical downsampling and processing of initial features. Instead of using absolute point coordinates, we input the distances $d_{ij}$ to the anchor points into the DGCNN, with dimensionality handled through linear layers of size $k$. This modification is applied consistently across the AdaPoinTr architecture, ensuring encoding with the same anchor points throughout the network.

The use of distances also affects the training objective. The original AdaPoinTr loss function focuses on predicting the correct coordinates for noisy points perturbed during training. Since our network predicts distances, we modified the loss function to account for the noisy distances between noisy input points and noise-free anchor points. This ensures that our network can still remove noise from the inputs, even when operating in distance space.

Similarly, in the self-attention layer, we replaced the point coordinates with the distances to anchor points to better capture the geometric relationships in the point cloud while maintaining invariance to input rotation.

These changes rely on the intuition that distances can effectively serve as encoded point coordinates. Specifically, two points will remain neighbors in Euclidean space when their distances to anchor points are used as descriptors, effectively representing the points in a "distance space." Therefore, point coordinates can be replaced by distances to anchor points without losing the geometric relationships between the points.

The final output distances are inherently unaffected by any rotations applied to the input, ensuring both consistency and rotation invariance. To reconstruct the completed point cloud with accurate 3D geometry, we solve an optimization problem to determine the final coordinates of the 3D shape from the predicted distances.

### 3.3 POINT COORDINATE OPTIMIZATION

To finalize the point cloud completion, the predicted distances must be converted into point coordinates. This step involves finding the coordinates of points whose distances to known anchor points match the predicted values. More formally, the optimization problem is defined as:

$$\min_{x,y,z} \sum_{i=1}^{k} \left( \sqrt{(x - x_{a_i})^2 + (y - y_{a_i})^2 + (z - z_{a_i})^2} - D_c(i) \right)^2, \tag{1}$$

where $x_{a_i}, y_{a_i}, z_{a_i}$ are the coordinates of the known anchor points, and $x, y, z$ are the coordinates of the point to be determined, while $D_c$ represents the distances predicted by the network. This optimization problem is solved independently for each point using the Levenberg-Marquardt algorithm (19) to recover the full object shape.

The final coordinates predicted by the pipeline retain the same orientation as the partial input cloud, ensuring that the method is rotation-equivariant. Moreover, due to the model's rotation invariance, the predicted complete object coordinates remain consistent under varying transformations of the input.

## 4 EXPERIMENTS

We conduct point cloud completion experiments on different datasets with different input transformations to evaluate our method's effectiveness and robustness over varying input conditions. These experiments are performed on established benchmarks: PCN(43) and KITTI(7). We also introduce additional evaluation on OmniObject(34) dataset to demonstrate generalizability and robustness on real-world object scans. Results demonstrate the effectiveness of our approach across many datasets, showcasing superior performance compared to existing models when presented with nonrotated or non-canonical inputs.

For all of our experiments, we refrained from applying rotations during training and evaluated our models on two conditions: no rotation and rotation across three dimensions. Only in OmniObject experiments did we not rotate the input point clouds and test the methods' capability of handling arbitrary rotations originating from the projection of the depth maps with unknown extrinsic parameters.

### 4.1 TRAINING SETUP

We used the Pytorch framework and the Adam optimizer for our implementation with $\beta_1 = 0.9$ and $\beta_2 = 0.999$. We initialized the learning rate at 0.001 and utilized a learning rate scheduler that multiplied the learning rate by 0.98 in every 15 epochs. We train our models until the validation loss does not improve over the last 30 epochs and a maximum of 200 epochs. We utilized the Chamfer Distance using the L1 distance in our loss function, which is calculated as:

$$CD(P, Q) = \frac{1}{|P|} \sum_{p \in P} \min_{q \in Q} \|p - q\|_1 + \frac{1}{|Q|} \sum_{q \in Q} \min_{p \in P} \|q - p\|_1$$

Where $P$ and $Q$ are the two point sets and $\| \cdot \|_1$ denotes the L1-norm. To construct our loss function on predicted distances as:

$$L = CD(\hat{D}_c, D_c(P, A))$$

where $\hat{D}_c$ is the predicted distances from the completed point cloud points to the anchor points and $D_c(P, A)$ is the ground truth distances between points in the complete shape and anchor points.

### 4.2 THE PCN BENCHMARK

**The PCN Dataset** consists of 8 categories derived from the ShapeNet dataset and includes numerous instances of complete and partial point clouds. Partial clouds are derived from complete clouds by back-projecting depth images from 8 different viewpoints with varying numbers of points. Adhering to established conventions, we upsampled/downsampled 2048 points from the input to construct our inputs and generate 16,384 points as the final output.

Following the existing methods, we used the PCN dataset to evaluate the models' shape completion capabilities. Additionally, we generated rotated inputs from the same dataset to assess the performance degradation of the models under rotation. To generate them, we selected random degrees between 0-180 for all three axes and applied these rotations simultaneously. When evaluating existing methods, the rotation is applied to the ground truth complete shapes to align them with the final predicted complete shapes.

**Evaluation.** We used Chamfer Distance under the L1-norm as both the loss function and evaluation metric for point cloud completion. It measures the distance between two unordered sets and is commonly used as a metric for PCN dataset benchmark. Following existing methods, we used the same train/validation/test splits of the PCN dataset and reported the value of the Chamfer Distance with L1 norm, multiplied by 1000. Table: 1 shows the detailed results, and Figure 3 depicts qualitative comparison with other methods.

Our method is the only model where prediction is unaffected by the input rotation and achieves competitive results on unrotated inputs. As shown in Figure 3, ESCAPE can achieve high-resolution outputs for planes and cars that fall under shape categories without variation. However, it may struggle to capture fine details in categories with diverse geometries, as demonstrated by the sofa example in Figure: 3.

Rotating or shifting the coordinates in the inputs of existing methods leads to significant performance degradation in completion. Their predictions are subject to high noise and structural deformations. As shown in Figure 3, most predictions become barely recognizable when the input is rotated. Interestingly, Table1 reveals that models with better CD-L1 scores on inputs without rotation achieve worse when the input is rotated. This phenomenon is that existing methods overfit the dataset they trained on and, therefore, lack robustness against real-world scenarios. On the other hand, ESCAPE can yield identical predictions under rotation, superior to all other methods when the inputs are perturbed with rotation, and perform similarly when the input rotation is not present. This makes ESCAPE applicable to scenarios that require non-canonical point processing.

| | Snowflake(35) | | Seedformer(46) | | PointTr(41) | | AdaPoinTr(42) | | AnchorFormer(2) | | Ours | |
|---|---|---|---|---|---|---|---|---|---|---|---|---|
| | normal | rotated | normal | rotated | normal | rotated | normal | rotated | normal | rotated | normal | rotated |
| Plane | 4.29 | 16.80 | 3.85 | 12.89 | 4.75 | 15.73 | **3.68** | 14.02 | 3.70 | 14.53 | 7.2 | **7.2** |
| Cabin | 9.16 | 32.70 | 9.05 | 34.49 | 10.47 | 31.63 | **8.82** | 41.39 | 8.94 | 40.27 | 14.5 | **14.5** |
| Car | 8.08 | 21.16 | 8.06 | 21.51 | 8.68 | 21.21 | **7.47** | 20.85 | 7.57 | 22.86 | 10.1 | **10.1** |
| Chair | 7.89 | 27.45 | 7.06 | 26.97 | 9.39 | 26.46 | **6.85** | 35.45 | 7.05 | 35.80 | 11.5 | **11.5** |
| Lamp | 6.07 | 22.55 | 5.21 | 24.78 | 7.75 | 18.76 | 5.47 | 20.35 | **5.21** | 24.14 | 8.1 | **8.1** |
| Sofa | 9.23 | 28.55 | 8.85 | 30.58 | 10.93 | 29.41 | **8.35** | 35.70 | 8.40 | 37.24 | 14.2 | **14.2** |
| Table | 6.55 | 37.08 | 6.05 | 35.91 | 7.78 | 33.84 | **5.80** | 51.06 | 6.03 | 50.66 | 9.4 | **9.4** |
| Boat | 6.40 | 18.59 | 5.85 | 17.01 | 7.29 | 17.36 | **5.76** | 15.71 | 5.81 | 18.00 | 9.3 | **9.3** |
| Avg | 7.21 | 25.61 | 6.74 | 25.52 | 8.38 | 24.30 | **6.53** | 29.32 | 6.59 | 30.44 | 10.5 | **10.5** |

Table 1: Results on PCN dataset. We use CD-L1($\times 1000$) as an evaluation metric and report the results for normal and rotated inputs. The best results for both input types are written with bold letters

### 4.3 OMNIOBJECT3D BENCHMARK

Our next experiment focuses on a real-world scenario that requires the reconstruction of real-world objects from depth maps whose pose parameters are unknown. To create the experiment, we collected 7 categories from the OmniObject(34) dataset that are also included in the PCN(43) dataset. Each category has multiple samples with a complete point cloud and 100 depth maps from different viewpoints. We back-projected these depth maps to obtain partial point clouds with unknown orientations. Before feeding the models with the inputs, we applied normalization on partial point clouds to match their distribution with the PCN dataset. We applied the inverse of the input transformation to the complete point clouds to be able to evaluate the completion performance.

We observed that the OmniObject dataset is challenging for point cloud completion due to two main reasons. First, it contains objects with varying geometry and dissimilar to their PCN pairs. Secondly, incorrect depth and intrinsic parameters for some samples lead to isolated points in the input point clouds, affecting the method's completion ability significantly. We observed that the mentioned challenges cause the metric value to degrade, misleading the performances of the models. Therefore, to remove these outliers, we report the median per category instead of the average of each sample in each category.

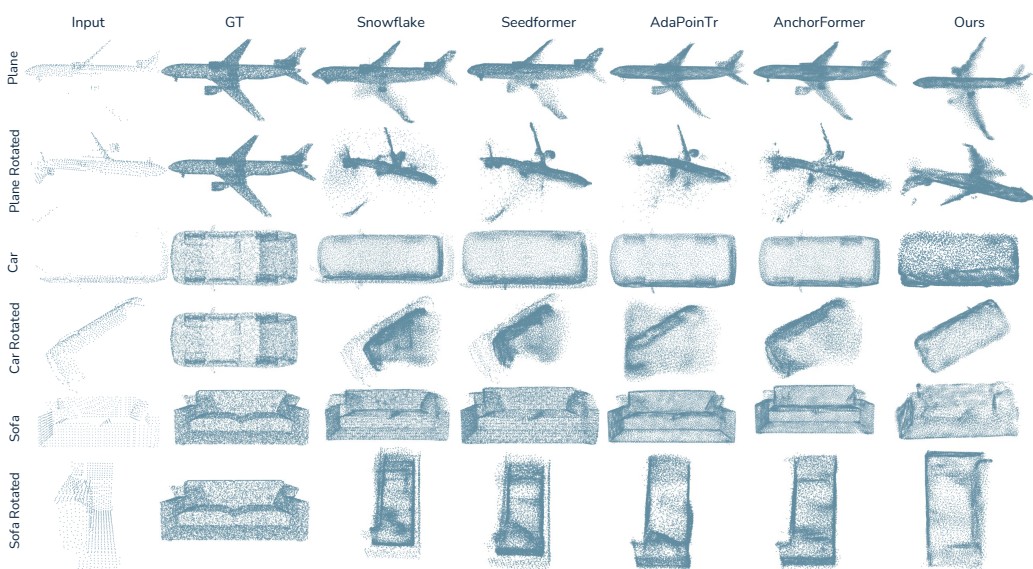

Figure 3: Qualitative comparison of models trained on PCN dataset and tested with and without rotated inputs. Each row contains the input to the model in its first column. Every even row contains the rotated input of the preceding row.

|        | Snowflake(35) | Seedformer(46) | PoinTr(41) | AdaPoinTr(42) | AnchorFormer(2) | Ours     |
|--------|---------------|----------------|------------|---------------|-----------------|----------|
| Plane  | 16.36         | **13.90**      | 15.49      | 17.53         | 18.20           | 16.00    |
| Cabin  | 37.45         | 34.61          | 46.13      | 54.02         | 51.00           | **12.09** |
| Car    | 34.38         | 32.70          | 40.07      | 49.97         | 47.69           | **26.3** |
| Chair  | 23.12         | 21.08          | 23.12      | 31.74         | 27.84           | **7.2**  |
| Lamp   | 54.78         | 59.02          | 54.78      | 67.36         | 72.13           | **49.85** |
| Sofa   | 28.36         | 26.16          | 28.36      | 38.88         | 38.42           | **9.7**  |
| Boat   | 21.71         | 19.47          | 22.88      | 22.49         | 25.60           | **10.62** |
| Avg    | 31.14         | 29.56          | 32.97      | 39.57         | 40.12           | **18.82** |

Table 2: Results on OmniObject dataset. We report median CD-L1($\times$1000) per category as an evaluation metric and report the average of all categories. The best results for both input types is written with bold letters

Similar to the PCN dataset, our method stands as the only model not affected by the input rotation. Results in Table 2 show that it is capable of handling unknown object poses and still able to reconstruct objects in high resolution. Qualitative results in Figure 4 is another evidence of our method's capability of handling arbitrary poses while existing methods failed to generate a structured geometry. We refer readers to supplementary material for qualitative results.

## 4.4 KITTI CARS BENCHMARK

Another real-world scenario experiment includes predicting complete car object geometries in the KITTI dataset, which contains incomplete point clouds from LiDAR scans in real-world scenes. Following the existing literature, we pretrain our model only with car images from the PCN dataset. As the complete point clouds are not available in this dataset, we reported the MMD and Fidelity score under two different setups: (i) original partial points and (ii) inputs with a random rotation around a single axis to mimic the movement of a vehicle. Table 3 shows the performance of the models.

The MMID results in the rotated samples show that existing methods fail to complete the cars when a simple yet realistic rotation is applied to the inputs. Moreover, our method achieved similar scores to other methods in inputs without rotation while training without any augmentation techniques.

Figure 4: Qualitative comparison of models trained on PCN dataset and tested on OmniObject dataset. Each row contains the input to the model in its first column.

| Methods | Fidelity | | MMID | |
|---|---|---|---|---|
| | normal | rotated | normal | rotated |
| PoinTr(41) | **0.0** | **0.0** | **4.6** | 6.15 |
| Snowflake(35) | 1.3 | 1.77 | 7.5 | 16.08 |
| Ours | 1.81 | 1.81 | 5.0 | **5.93** |

Table 3: Results on KITTI Cars dataset for normal and rotated inputs. Fidelity and MMID metrics are calculated using CD-L2 (x1000) distance.

## 4.5 ABLATION STUDIES

### 4.5.1 OTHER ENCODINGS

Shape completion literature lacks robust methods against input transformations to be a baseline for our process. To fill this gap and provide rotation equivariance encodings as a baseline to our method, we proposed three methods: (i) We adopt equivariant DGCNN architecture proposed by Vector Neurons (3) for part segmentation task with upsample followed by a MLP to generate complete shapes in different scales. (ii) We modified the DGCNN architecture built with Vector Neuron layers for point cloud completion by combining its rotation-invariant encoder with the decoder of the Snowflake. Referred as Vector Neurons in Table 4. (iii) Snowflake(35) network processing point pair features(PPFs) and modified to be rotation-invariant. Referred as PPF-Snowflake in Table 4. Results in Table 4 shows that our method surpassed the baseline in all categories, showing superior point cloud completion.

| Method | Plane | Cabin | Car | Chair | Lamp | Sofa | Ship | Avg |
|---|---|---|---|---|---|---|---|---|
| DGCNN (Vector Neurons)(3) | 93.66 | 93.66 | 93.66 | 93.66 | 93.66 | 93.66 | 93.66 | 93.66 |
| Snowflake (Vector Neurons)(3) | 10.65 | 26.64 | 11.92 | 17.86 | 22.82 | 22.90 | 18.19 | 18.62 |
| Snowflake (PPF) | 8.68 | 19.68 | 10.95 | 19.36 | 25.96 | 18.62 | 16.07 | 17.46 |
| Ours | **7.2** | **14.5** | **10.1** | **11.5** | **8.1** | **14.2** | **9.3** | **10.5** |

Table 4: Comparison of our feature extractor with Vector Neurons to generate a global feature vector for the same decoder module.

### 4.5.2 Anchor Point Selection

As part of our ablation study, we further investigate our anchor point selection algorithm and analyze the characteristics of optimal point sets. We evaluated the performance of different anchor point selection methods on a subset of the PCN dataset. The results are presented in Table 5. For visual reference, we refer the reader to the supplementary material, which illustrates the points selected by the algorithms.

| Algorithm | Threshold/Radius | CD-L1 Score |
|---|---|---|
| *Clustering* | 0.0 | 14.80 |
| *Clustering* | 0.5 | 15.73 |
| *Ball Query* | 0.05 | 14.23 |
| *Ball Query* | 0.075 | **13.58** |
| *Ball Query* | 0.1 | 15.00 |
| *Ball Query* | 0.15 | 15.76 |
| *FPS* | - | 14.74 |

Table 5: Results of different anchor point selection algorithms on a subset of the PCN dataset. CD-L1 (x1000) distance is reported.

In these experiments, we also tested an alternative algorithm to select anchor points based on the curvature values of points. First, we use the FPS algorithm to select $a$ anchor points, then use these points as cluster centers, assigning all other points to their closest cluster. For each cluster, we select the point with the maximum curvature as the anchor point, provided its curvature exceeds a predefined threshold. If no point in the cluster meets the curvature threshold, we retain the original FPS-selected point as the anchor. The results of this algorithm, referred to as "Clustering," are shown in Table 5.

For comparison, the final row of Table 5, labeled "FPS," shows the results of using the FPS algorithm alone without further refinement. Both results demonstrate the importance of well-distributed anchor points around the shape for the model's success. Selecting anchor points from clusters provides more flexibility for points to move away from their initial positions, thereby covering less area of the object's geometry.

Additionally, we limit the refinement of anchor points to a maximum distance (radius) from their initial positions. The rows labeled "Ball Query" in Table 5 show the results for different radius values. The results indicate that a larger radius does not improve performance, as it allows too much flexibility for anchor points to shift around the shape, resulting in less coverage of critical regions. This observation aligns with the conclusions from previous experiments.

The "Ball Query" results further show that selecting points with respect to their curvature is beneficial. By allowing points to move a small distance, such as 0.075, from their initial positions, we were able to choose better anchor points, resulting in improved point cloud completion performance.

## 5 Conclusion

In this work, we presented Equivariant Shape Completion via Anchor Point Encoding (ESCAPE), a novel method for achieving rotation-equivariant shape completion. Our approach tackles the key challenge of reconstructing object geometries from various orientations by leveraging rotation-equivariant keypoint detection and a distance-based feature encoding inspired by the D2 shape distribution.

Through comprehensive experiments on the PCN, KITTI, and OmniObject datasets, ESCAPE consistently outperformed existing models, mainly when dealing with rotated input data. This demonstrates the method's robustness in scenarios where input orientation is not controlled or known in advance.

ESCAPE provides a practical solution for dynamic environments, such as robotic manipulation and real-time object recognition, by enabling effective shape completion without requiring prior knowledge of object orientation or additional pose estimation modules. The combination of rotation-equivariant keypoint detection and a transformer-based architecture enables ESCAPE to reconstruct

objects with high precision, capturing fine details and maintaining geometric consistency throughout the process.

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
