# ESCAPE: Supplementary material

## .1 Visualization of KITTI Experiments

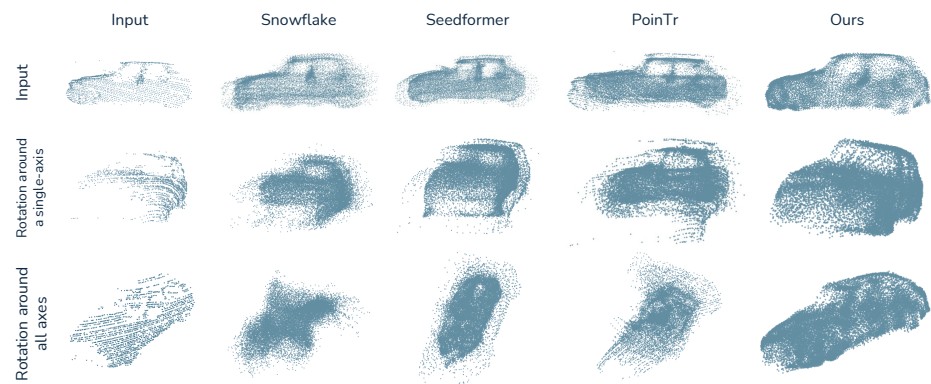

Figure 1: Qualitative comparison of models fine-tuned on PCN dataset cars category and tested on KITTI dataset. The first row contains the original input. The second row contains a single-axis rotation of the input mimicking the movement of a car. The final row contains the partial input rotated in all three axes.

## .2 Visualization of point curvatures

Figure 2 illustrates the calculated curvature value for each point in the input and derived refined anchor points using the mentioned algorithm. The figure depicts that our algorithm can select points with high curvature and still cover all regions of the shape, hence good anchor points for point cloud completion.

## .3 Visualization of point curvatures

Outputs of using different anchor point selection algorithms are given in Figure 3. The visualizations show that starting with FPS generates good candidates as the points picked from the algorithm covers all parts of the shape. Furthermore, these points can be refined by utilizing their curvature values (Figure 2 for curvature heatmap and bottom right image in Figure 3) yet the refinement should be limited. It is observed in the Figure 3, not limiting the refinement can end up points closer to each other(top right and bottom left images) which might end up with smaller anchor points (bottom left). Therefore, Figure 3 shows that a good balance for a point between curvature and being able to cover parts is achieved with our proposed algorithm.

## A Limitations

While our approach demonstrates significant advantages in achieving rotation-equivariant shape completion, it has limitations. Firstly, although our method is less data-driven than techniques that learn rotation through augmentation, it still requires substantial training data to achieve high performance. This dependency on data can be a bottleneck, especially for applications where labeled data is scarce or expensive to obtain.

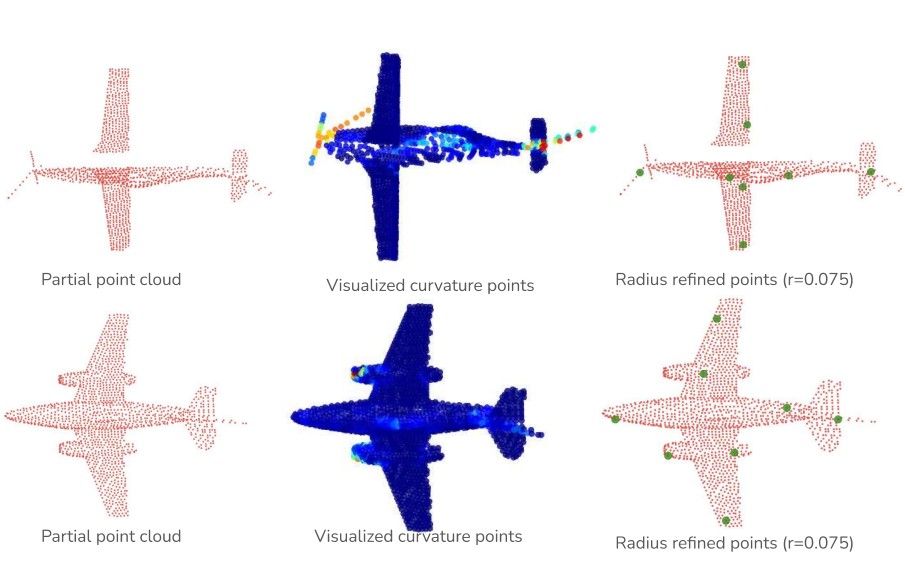

Figure 2: Heatmap of the point curvature values and derived anchor points

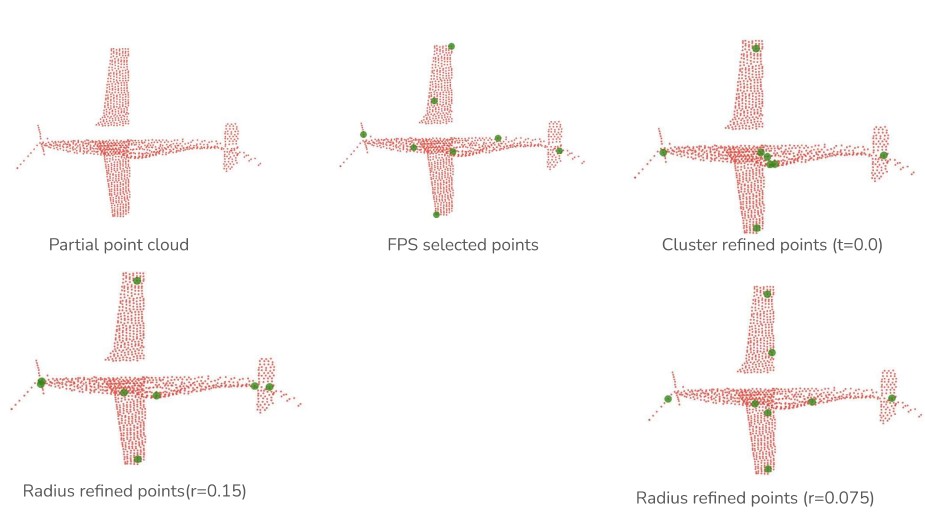

Figure 3: Comparison of different anchor point selection algorithms

Another key limitation of our approach is the optimization procedure to find the coordinates of the complete shape. This procedure prevents the model from being rotation-invariant, which will limit its applicability to some real-world applications.

Furthermore, while distance-based encoding contributes to rotation invariance, it also introduces additional complexity to the learning process. This complexity can sometimes result in lower

performance on standard, non-rotated cases. In scenarios where the objects are consistently presented in a canonical alignment, methods that memorize these aligned shapes may outperform our approach. The trade-off between achieving rotation invariance and maintaining high performance on canonical shapes is an essential consideration for the practical deployment of our model.