# OpenReview forum: "ESCAPE: Equivariant Shape Completion via Anchor Point Encoding"
_ICLR.cc/2025/Conference — ICLR 2025 Conference Withdrawn Submission_

### Official Review · Reviewer_pSKb · 2024-11-03

**Soundness:** 3
**Presentation:** 1
**Contribution:** 2
**Rating:** 3
**Confidence:** 3

**Summary:**

ESCAPE proposes to use distances to anchor points as features that represent point clouds. The anchor points are selected by curvature in local clusters. Point cloud completion is performed by an accordingly modified AdaPoinTr in the anchor-distance-space. The positions of the points in Cartesian space are optimized from the over determined distance representations. The different representation is orientation agnostic, thus the trained model will be as well.

**Strengths:**

Simple orientation agnostic point cloud representation.

**Weaknesses:**

* The Transformer-Architecture is not described in the paper. It is necessary to rely on the descriptions of the AdaPoinTr and FoldNet papers to understand what is happening. I'd like to see a method section where first the things that are built upon are described concisely and then the exact changes are highlighted. This would allow the reader to see what is new and what has been around already.
* As far as I can see the only new idea is to represent the input and output point clouds in the distance space. (little novelty)
* Evaluation shows method performs worse than reference methods on oriented inputs.

**Questions:**

* Please describe the architecture. Given a input point cloud of shape [1, N_in ,3], what exactly are all the steps that are taken to get [1, N_out, 3] in the end?
* Cite the Shape Distributions paper by Robert Osada
* Add Flex Convolution (Million-Scale Point-Cloud Learning Beyond Grid-Worlds) to related works 2.1.
* Add numbers to all equations (CD and Loss).
* Table captions are usually put on top of tables.
* The results indicate, that other methods (e.g. AdaPointTr) perform better if the orientation is known. Why not use a pose estimation with the original AdaPointTr?
* You claim to use point clouds to get higher resolution results compared to alternative representations such as SDFs. I'd like to know what exact resolutions of point clouds you trained on and evaluated on (N). How many anchor points (M) do you use?
* What is the training time for the 30 epochs, what hardware did you use and how much memory does the training need?
* What is the inference time (from point partial point cloud going in to complete point cloud coming out)?

---

### Official Review · Reviewer_augc · 2024-11-03

**Soundness:** 2
**Presentation:** 2
**Contribution:** 1
**Rating:** 3
**Confidence:** 4

**Summary:**

The paper introduces an architecture designed for the task of 3D shape completion, with a focus on achieving robustness through end-to-end rotation-equivariance. This is accomplished by sampling a set of anchor points, and representing the shape points using the relstive distances to those anchor points. The model proceeds with a AdaPointTr applied to the equivariant representation for shape completion.

**Strengths:**

+ The paper addresses the impactful and challenging problem of 3D shape completion, under unknown orientation.
+ The proposed method is straightforward and easy to implement.
+ Compared to the baseline, the results are robust to rotation without requiring augmentation during training.
+ Introduces a rotation-invariant adaptation of AdaPointTr.
+ Proposes an anchor selection mechanism based on point curvature in the point cloud.

**Weaknesses:**

**Weaknesses:**
- The claim that this is the first rotation-equivariant shape completion method is exaggerated. Other rotation-equivariant shape completion methods exist, such as [1], and [2] and [3] can also be easily adapted for shape completion. These methods should be addressed in the related work, and comparisons with [1] are necessary.
- The usage of curvature for anchor point selection lacks motivation.
- Given a partial point cloud, with large missing regions, such as a half aiplane; it seems no anchor points will be chosen in the vicinity of the missing region causing less accurate reconstruction. Is that the casue for the blurry completion?
- The main contribution of the method is the equivariant representation. Given this, I am interested in understanding how this representation performs on tasks beyond shape completion. For example, how would it perform on point cloud classification? It seems feasible to use the encoder alone, without the decoder, for classification purposes. Have you tested the performance of this equivariant representation on such tasks, and if so, how does it compare to other representations?
- Vector Neurons [2] is mentioned in the ablation study but lack references in both the related work and results sections, even though they can be used for equivariant shape completion.
- Were methods in comparison allowed to train with augmentations? Section 4 states, “For all of our experiments, we refrained from applying rotations during training and evaluated our models on two conditions: no rotation and rotation across three dimensions.” However, these baseline models were designed to be trained with rotation augmentations. Please include comparisons with methods trained as intended.
- The proposed method is not provably equivariant duo to the randomness of the FPS. Due to this randomness, the anchor selection process may yield different sets of anchor points on each run, potentially making the method non-equivariant, as different point cloud rotations could lead to varied anchor selections. This issue should be addressed.

**Missing References:**
- [1] H. Wu and Y. Miao, "SO(3) Rotation Equivariant Point Cloud Completion using Attention-based Vector Neurons," 2022 International Conference on 3D Vision (3DV), Prague, Czech Republic, 2022, pp. 280-290, doi: 10.1109/3DV57658.2022.00040.
- [2] Deng, Congyue, et al. "Vector neurons: A general framework for so (3)-equivariant networks." Proceedings of the IEEE/CVF International Conference on Computer Vision. 2021.
- [3] Atzmon, Matan, et al. "Frame averaging for equivariant shape space learning." Proceedings of the IEEE/CVF Conference on Computer Vision and Pattern Recognition. 2022.
- [4] Yu, Xumin, et al. "Pointr: Diverse point cloud completion with geometry-aware transformers." Proceedings of the IEEE/CVF international conference on computer vision. 2021.

**Questions:**

- In Section 3.1, "Anchor Points Encoding," the anchor selection process is described as "Clustering," which matches the method described in Section 4.5.2, "Anchor Point Selection." However, the best-performing method, according to the results, is "Ball Query." Do the authors use "Clustering" in practice? If so, why not use "FPS," given that the CD-L1 score for FPS is lower than "Clustering"?

- In line 230, the authors present an equation regarding the Laplacian of the normal vectors of points. I have two questions regarding this:
  1. Does the method require that the input point cloud includes normal vectors?
  2. it is not specified how the calculated Laplacian is used?

- In Table 1, results for your method are identical between normal and rotated cases, which is unexpected given the partial equivariance due to the randomness in the FPS algorithm. This contrasts with Table 3, where normal and rotated cases differ. Could you explain this inconsistency?

- In Table 4, all results for DGCNN are identical which seems unreasonable. Please specify the dataset and metric used for evaluation in this table.

---

### Official Review · Reviewer_qqv2 · 2024-11-04

**Soundness:** 3
**Presentation:** 2
**Contribution:** 4
**Rating:** 5
**Confidence:** 4

**Summary:**

This work investigates the robustness of currently used point cloud completion methods and concludes that their performance is heavily dependent on the pose of the input point clouds. To address this issue, it proposes a novel framework for point cloud completion that is equivariant to the pose of the input point clouds. Specifically, it introduces an invariant point encoding that encodes the position of each input point by computing its distance with respect to a set of equivariantly selected anchor points. This invariant encoding is then passed to a transformer architecture that outputs the relative distances of points in the predicted point cloud, again with respect to the same set of anchor points. Given these relative distances, the final position of the predicted points can be inferred by solving an optimization problem. The authors evaluate their method on both synthetic and real-world point cloud completion benchmarks, showing improvements over state-of-the-art non-equivariant methods in cases where the input point clouds have random poses.

**Strengths:**

- The proposed invariant point cloud encoding can be easily used to modify current non-equivariant architectures into equivariant ones while retaining the expressivity required for the challenging task of point-cloud completion.
- The experimental results highlight both the limitations of non-equivariant point-cloud completion methods when dealing with randomly rotated input-point clouds, as well as the benefits of utilizing the proposed equivariant alternative.

**Weaknesses:**

- This work does not compare or reference previous works on equivariant point cloud completion, specifically the work:
[1] H. Wu and Y. Miao, "SO(3) Rotation Equivariant Point Cloud Completion using Attention-based Vector Neurons," 3DV (2022)
- In the experimental evaluation, the non-equivariant methods are trained without using rotations as data augmentations. Since data augmentations are a commonly used alternative to equivariant networks, excluding them during training makes it harder to compare the benefits of the proposed method over non-equivariant by-design alternatives.
- The presentation of the overall architecture and of the modifications made to the baseline AdaPointTr model is limited, making it harder for the reader to evaluate the proposed changes.

**Questions:**

- How does the proposed method relate and compare with the work [1] mentioned in the weaknesses section of the review?
- Why are the non-equivariant networks trained without rotation as data augmentations? Can the non-equivariant networks be made more robust to input rotations if we perform these augmentations?
- A more detailed description of the final architecture (inputs/outputs/layers), after the modifications described in Section 3.2, would benefit the completeness of this work.
- Minor: The content extends to the 11th page, while the submission page limit is 10 pages of content.

---

### Official Review · Reviewer_TUQK · 2024-11-11

**Soundness:** 2
**Presentation:** 3
**Contribution:** 2
**Rating:** 3
**Confidence:** 5

**Summary:**

**Summary**:
This paper deals with the task of equivariant shape completion from point clouds. First the authors preprocess the point clouds to select anchor points of high curvature. Then they encode the partial point cloud as distances w.r.t. these selected anchors. A transformer based architecture on the distance space selects the final completed point cloud in distance space. The authors then perform an optimization to localize the point from the distances. The experiments show that in some case the authors perform better shape completions.

**Strengths:**

**Significance**:
1. Point cloud completion is a significant task with extended literature.
2.  The authors show cases where the shape completion is better than some compared methods.

**Novelty**:
1. **High curvature anchor point selection**: The authors's approach to select the anchor points is novel to my knowledge.
2. **Distance Representation**: The reduction of absolute coordinates to distances w.r.t. invariantly selected anchor points is novel to my knowledge.

**Clarity**: The paper is clearly written. Some questions in the end of the review.

**Correctness**:  Strength The paper is in general correct. Some questions regarding possible pitfalls of the method discussed next.

**Weaknesses:**

**Significance**:
1. For the authors's approach significance to be evaluated I would require a more extended discussion in the limitations about the sparsity of the point clouds and other failure cases of the method.
2. "Proper Experimental evaluation missing": The performance of the method by a simple adaptation of the base network is much worse in the canonical frame. The authors have not properly compared performance with 1) PCA canonicalization of the rotated point clouds 2) noisy versions of point clouds

**Novelty**:

1. **A lot of Missing Literature on Equivariant Shape Completion**: The paper in lines 064-068 claims that "these advancements (rotation-invariant descriptors) have yet to be fully extended to shape completion... but do not directly facilitate the reconstruction of entire object geometry in a rotation-equivariant manner". Also in line 087 "first end-to-end rotation-equivariant shape completion method..."

These paragraphs omit a lot of papers in the literature on equivariant shape reconstruction and completion from point clouds.

  - On reconstruction:

    1. SE(3)-Equivariant Attention Networks for Shape Reconstruction in Function Space, ICLR 2023.
    2. 3D Equivariant Graph Implicit Functions, ECCV 2022.

  - On Shape completion:
    1. SCARP: 3D Shape Completion in ARbitrary Poses for Improved Grasping, ICRA 2023.
    2. SO(3) Rotation Equivariant Point Cloud Completion using Attention-based Vector Neurons 3DV 2022.

Incorporating and discussing those papers properly in the related work would make the novelty of the paper clearer with respect to approaches that have already been published in the literature on the topic. Right now, the novelty w.r.t. other equivariant shape reconstruction/completion methods that also use transformers is obscured.

**Correctness**
1.  **Degeneracy**: When does the optimization problem (1) have a unique solution? If it has more solutions what does that mean for the absolute coordinate of a point? How does this relate to the anchor selection and in turn how does it relate to how incomplete the initial point cloud is?
2.  **Missing comparisons/metrics**: It is usual in the equivariant literature (see missing papers above) to evaluate also on the case where the non-equivariant counterpart is presented with rotation augmentations during training.
3. **Notation** Line 309 CD is described between point clouds while line 314 CD is used with distance inputs. Also Line 228 $m$ is used for the number of points in a cluster while in figure for the total points in completed point cloud.

**Reproducibility**:
1. **Missing Key Details**: Some more details on how the self-attention is altered, how the DGCNN is altered and how the final number $m$ of points is generated would be crucial for understanding. The paper is not self contained right now.
2. **Missing code** Parts of the pipeline regarding alternations of DGCNN, the self-attention layer and the loss hinder reproducibility. There is no code to accompany the manuscript that would help on that front.

**Questions:**

Questions and suggestions:
1. Have the authors tested standard non-equivariant methods on points clouds that have been canonicalized by PCA? This is a naive baseline comparison that would validate the method's performance.
2. The way the high curvature points are computed, ie. using the smallest eigenvalue of the covariance matrix seems pretty susceptible to noise. Have the authors tried to do an ablation when the point clouds are corrupted with noise?
3. How much time does the processing of creating neighborhoods, normals and choosing the anchors of high curvature take w.r.t. the forward pass?
4. How much time does the optimization problem take w.r.t. the forward pass? Is is always converging?
5. Ablation with random FPS points as anchors instead of high-curvature anchors is not provided, to strengthen the arguments of the anchor significance that is the main novelty of the paper.

---

### Note · Authors · 2024-11-15

I have read and agree with the venue's withdrawal policy on behalf of myself and my co-authors.